# Investigation of Electromagnetic Pulse Compaction on Conducting Graphene/PEKK Composite Powder

**DOI:** 10.3390/ma14030636

**Published:** 2021-01-30

**Authors:** Quanbin Wang, Deli Jia, Xiaohan Pei, Xuelian Wu, Fan Xu, Huixiong Wang, Minghao Cao, Haidong Chen

**Affiliations:** 1Department of Oil & Gas Production Equipment, Research Institute of Petroleum Exploration and Development, Xueyuan Road 20#, Beijing 100083, China; wqb_upc@petrochina.com.cn (Q.W.); jiadeli422@petrochina.com.cn (D.J.); peixh@petrochina.com.cn (X.P.); 2School of Mechanical Engineering, Jiangsu University, Xuefu Road 301#, Zhenjiang 212000, China; 2211903042@stmail.ujs.edu.cn (H.W.); 18356635336@163.com (M.C.); chd0527@163.com (H.C.)

**Keywords:** graphene composite, conductivity, spark plasma sintering, magnetic force, electromagnetic pulse compaction

## Abstract

Polymer-composite materials have the characteristics of light weight, high load, corrosion resistance, heat resistance, and high oil resistance. In particular, graphene composite has better electrical conductivity and mechanical performance. However, the raw materials of graphene composite are processed into semi-finished products, directly affecting their performance and service life. The electromagnetic pulse compaction was initially studied to get the product Graphene/PEKK composite powder. Simultaneously, spark plasma sintering was used to get the bars to determine the electrical conductivity of Graphene/PEKK composite. On the basis of this result, conducting Graphene/PEKK composite powder can be processed by electromagnetic pulse compaction. Finite element numerical analysis was used to obtain process parameters during the electromagnetic pulse compaction. The results show that discharge voltage and discharge capacitance influence on the magnetic force, which is a main moulding factor affecting stress, strain and density distribution on the specimen during electromagnetic pulse compaction in a few microseconds.

## 1. Introduction

With the rapid development of automobile, aviation, aerospace, and rail transit, lightweight materials have become a relatively important topic as an effective measure to pursue global energy conservation and environmental protection [1,2,3]. Polymer composite materials have the characteristics of light weight, high strength, corrosion resistance, fatigue resistance, and strong designability. Accordingly, they are extensively used in the automobile, aviation, and aerospace industries, among other fields [1,2,3,4,5]. Indeed, composite materials have become essential in the manufacture of automobiles and aircraft parts [5,6,7]. Boeing and Airbus use thermoplastic composites as load-bearing structural components, such as stiffeners, corner pieces, cabin seats and other secondary load-bearing structural members, wing leading edges, fuselage stiffened panels, flat wing torque boxes, and other main load-bearing structure transitions [4,7]. In the aerospace field, aromatic hydrocarbon polymers are the preferred composite materials, especially poly (ether-ketone-ketone) (PEKK) and poly (ether -ether -ketone) (PEEK), because of their excellent mechanical properties.

PEEK composite is a lightweight material that can bear the main load-bearing structure of an aircraft. As a torsion box with a span of 12 m, carbon fibre with 6 m length and 28 mm thickness can be realized by fibre-reinforcement technology to strengthen the upper beam of a PEEK engine pylon [7,8,9]. Applications of thermoplastic composites in aviation include the F/A18 fighter aircraft skin and AS4/PEEK composite material, which undergo re-melting forming preparation. The Boeing company uses GF/PEEK composite material to prepare Boeing 757 engine fairing. The weight of IM6/PEEK composite polymer material to prepare F-5F landing-gear inner and outer skin and observation platform has been reduced by 31% to 33% compared with an aluminium skin [8]. Generally, composite polymer materials of aromatic hydrocarbons are receiving extensive research attention and application.

In some harsh service environments, polymer-composite materials can replace traditional metal materials. They are an important means to solve the problem of corrosion and light weight. If polymer composites have high strength and conductivity, then they can replace metal materials for use in harsh environments. Graphene is known to be the thinnest and strongest material that can be separated from graphite materials [10,11,12]. Its fracture strength is 200 times higher than that of the best steel. At the same time, it has good elasticity, and the tensile range can reach 20% of its own size. Thus, graphene-based composites present many excellent properties [13,14]. Graphene can be used as an additive material or carrier to composite with PEEK [7,8,9]. Owing to its unique structure and properties, graphene has great value in improving the thermal, mechanical, and electrical properties of polymer-composite [10,12,15].

The conductive mechanism of conductive Graphene/PEEK polymer-composites is primarily through the formation of conductive passage [16]. Graphene conductive filler is introduced into the polymer matrix to form a binary composite material with good strength, toughness, and conductivity [17]. These excellent composites are extensively used in the fields of antistatic coatings, electromagnetic-shielding chemical sensors, thermal sensitive materials, and so on [16]. Polymer composites with conducting filler are extensively used in some fields, but numerous fillers are needed to obtain high conductivity, leading to difficulties in processing and embrittlement of materials. [18,19].

Graphene/PEKK composite materials are processed into semi-finished and finished products such as plates, bars, pipes, test parts, structural parts, and skin. To ensure the performance of Graphene/PEKK composite materials and to improve the quality of finished products, the moulding process is one of the most commonly used techniques. Graphene has been combined with the higher-performance thermoplastic PEKK to improve its electrical conductivity [20,21]. For effective applications and general acceptance of these materials, issues of effective cost and high mechanical performance must be addressed. The moulding method greatly influences the porosity of finished products. Many processing methods for thermoplastic-polymer powder materials are available. Increasing the moulding compaction density can improve the secondary processing performance of finished and semi-finished products. In addition to the dispersion of graphene during the preparation process, affecting mechanical properties and electrical conductivity, the moulding method directly affects its properties [22]. To fulfill the growing demand of high-performance conductive materials, researchers focused on the development of conductive polymer materials which can work at high temperatures and pressures, so the suitable pressure and temperature are key factors.

In general, the moulding process can be divided into hot and cold moulding. Hot moulding can improve the quality of products primarily by adjusting the temperature, whereas cold moulding refers to finished and semi-finished products obtained from powder at room temperature, and cold moulding mainly depends on pressure and pressure holding time to obtain specimens with high density, and it is easy to control the moulding size. Thermoforming includes hot pressing, extrusion, injection moulding, calendering, secondary moulding, and casting moulding. With the rapid development of computer technology, 3D-printing technology has been gradually applied in various fields [23,24,25]. It is a kind of rapid prototyping technology, also known as additive manufacturing, which is based on digital model files. It uses powdered metal or plastic and other bondable materials to construct by printing layer-by-layer object technology [26]. 3D-printing technology can be divided into melt deposition, selective laser sintering, selective laser-melting moulding and other types according to the different materials [26,27,28].

The use of 3D printing to form conductive Graphene/PEKK composite powder has certain problems. Firstly, the dispersion problem of graphene in the polymer matrix still needs to be solved to maximize its excellent mechanical properties and functionality [29]. Secondly, for a part of graphene, the printability of polymer matrix composites is insufficiently improved, and problems such as clogging of the nozzle and insufficient adhesion occur during printing. Thirdly, the types of polymers that can be used for 3D printing are limited [26,30,31]. Various materials are available for additive manufacturing (AM) composites, including synthetic and bio-based fibres and polymers. However, printable self-prepared Graphene/PEKK composite polymer powders in thermoplastic polymers with adequate viscosity remain limited. Theory and experiment need to be further improved and discussed, especially the scope of application for powder.

Some researchers have also used spark plasma sintering (SPS) to form specimens for polymer composite powder. SPS [32] is used to consolidate SiC-graphene nanoplatelet powders to form bulk samples. Reference [32] investigates the possibility of incorporating graphene nanoplatelets as a nanofiller in silicon carbide matrix composite to improve mechanical properties. Different processing conditions are used, including various temperatures and pressures, whilst limiting grain growth. The density and porosity of SiC-graphene nanoplatelet nanocomposites are determined as a function of processing temperature. Increasing the sintering temperature to 2100 °C results in a relative density of 90% with significantly improved mechanical properties.

The powder-moulding process at room temperature is called cold moulding. Previous research showed an important cold-moulding method [33] named low-voltage electromagnetic pulse compaction. It is applied to compact TiO_2_ and PZT powders in an indirect way. After selecting the appropriate processing parameters, TiO_2_ and PZT ceramics with higher density and better electrical properties than those produced by traditional static compaction are generated [34]. Low-voltage electromagnetic pulse compaction is concluded to be an efficient method of compacting ceramic powders, increasing the density of pressed parts and producing high-density sintered ceramics. Controlled enlargement of discharge voltage is another effective way to increase the density of ceramics. Moreover, the tools and coil are investigated in this paper.

B. Chelluri [35] described a novel rapid consolidation technique called dynamic magnetic compaction for consolidating advanced material powders. Powders of different materials such as intermetallics, refractory alloys, ceramics, nanomaterials, and composites can be fabricated into full-density parts by using dynamic magnetic consolidation (DMC). This high densification benefits the decrease in sintering temperatures, increase in strength, and improvement in microstructures with fine grain size [36]. Magnetic pulsed compaction (MPC) is a very short duration, high-density preform moulding method of producing bulks with varied consolidation conditions, followed by subsequent sintering. Density, hardness, and crack length show a gradual pattern or trend in magnetic pulsed compaction bulks that can yield almost fully dense, commercially applicable products with high mechanical properties after sintering. For metal powder, density is further improved up to 96% of the cast value by MPC at the maximum voltage. Uniform and fine microstructure formed in the alloy powders as atomised is almost maintained even after thermal MPC [37]. Some theoretical and experimental studies on the magnetic pulse compaction of nanosized powder have been carried out. G. Sh. Boltachev [38] analyzed the uniaxial pressing and radial compaction of powder using z-and θ-pinch setups. The objects of the study were two model systems involving alumina-based nanopowders. The value range was established where the most effective ”resonance” conditions were realized.

Few studies have focused on the moulding of Graphene/PEKK composites, especially the moulding of self-prepared Graphene/PEKK composites [39]. Identifying a general method suitable for the moulding process of self-prepared Graphene/PEKK composites is difficult. Therefore, a process plan suitable for the powder moulding of self-prepared conductive Graphene/PEKK composite materials is necessary to select, and the corresponding moulding process parameters need to be determined. In addition to theory and experiment, finite-element (FE) simulation analysis is an effective method for the preliminary exploration of new process schemes for self-prepared conductive Graphene/PEKK composite powders [40,41,42]. Some studies have been performed on the electromagnetic pulse compaction FE analysis method of metal and metal-composite ceramic powder [36,37,38], but almost no research has been conducted on electromagnetic pulse compaction of self-prepared conductive Graphene/PEKK composite powder.

In the present study, Graphene/PEKK composite powder was synthesized in the laboratory by way of in situ polymerization, and then it was called self-prepared Graphene/PEKK composite powder. Firstly, SPS was used to obtain a Graphene/PEKK composite compressed column with high electrical conductivity in order to determine the feasibility of electromagnetic pulse compaction conductive powder. Secondly, cold moulding-electromagnetic pulse compaction technology was primarily proposed to realize the moulding process of self-prepared Graphene/PEKK polymer powder at room temperature. In this paper, the finite element numerical analysis was used to determine process parameters during the electromagnetic pulse compaction moulding. The loose coupled FE analysis method with ANSYS and MSC.Marc was proposed, and it used ANSYS to obtain the magnetic force. Additionally, the magnetic-force distribution on different points was regarded as the boundary conditions of the MSC.Marc software. The FE numerical analysis method was initially used to determine the change law of process parameters during the cold moulding electromagnetic pulse compaction process and guided future theoretical and experimental research.

## 2. Performance of Graphene/PEKK Composite

Graphene used in this study was bought from Shenzhen Yuewang Energy Saving Technology Service Co. Ltd, Shenzhen, China. Terephthalyl chloride, diphenyl ether, aluminum trichloride (anhydrous), dimethylformamide, and dichloromethane were purchased from Sinopharmaceutical. The graphene was dispersed in dichloromethane, and then diphenyl ether, p-benzoyl chloride, N, N-dimethylformamide and anhydrous aluminum trichloride were added successively. After stirring evenly, the composite powder was obtained after cleaning, filtering, grinding and drying. According to the above technological process, Graphene/PEKK composite powder with 3.8% mass percentage of graphene was prepared. Among them, the molar ratio of diphenyl ether, terephthalyl chloride, N,N-dimethylformamide and aluminum trichloride is 1:1:2:6.

Thermal properties including thermogravimentric analysis (TGA) and differential scanning calorimetry (DSC) of self-prepared Graphene/PEKK composite powder were characterized before moulding of conductive Graphene/PEKK composite materials. The TGA test (TGA-NETZSCH STA 449F3, Selbu, Germany) was conducted from 50 to 1000 °C at a ramp rate of 10/min. The DSC test (DSC-NETZSCH STA 449F3, Selbu, Germany) was carried out between 0 and 450 °C at a heating/cooling rate of 10 °C/min.

Figure 1 shows the TGA curve of Graphene/PEKK composite with a grapheme mass content of 3.8%. As is shown in Figure 1, self-prepared Graphene/PEKK composite exhibits a one-step weight loss process, which indicates good physical compatibility between graphene and the PEKK matrix in the composite. A close look reveals that there was virtually almost no weight loss before 300 °C, and weight loss of Graphene/PEKK composite is around 5% when temperature increased to 450 °C, which can be defined as the start decomposition temperature. Main decomposition of the Graphene/PEKK composite occurs around 561 °C with a weight loss of 25%. It should be noted that the weight loss of the Graphene/PEKK composite is 45% when the temperature is around 1000 °C. The results indicated high thermal stability of the Graphene/PEKK composite.

Figure 2 presents DSC curves of Graphene/PEKK composite with a grapheme mass content of 3.8%. As shown in Figure 2, two types of transitions occur in this Graphene/PEKK composite during thermal cycling between 20 and 288 °C. The glass transition occurs mainly within the temperature range 114–129 °C, whereas the melting and crystallization transitions occur around at 261 and 203 °C upon heating and cooling, respectively. The glass transition temperature (*T_g_*) of this Graphene/PEKK composite may be defined as 125.5 °C. It should be noted that melting temperature and the corresponding viscosity of the polymer composite directly determines the difficulty and quality of flow forming process in engineering application. According to Figure 2, the melting temperature of Graphene/PEKK composite obtained from the DSC curve is 261 °C. Normally, the moulding temperature (sintering temperature) was initially determined between the glass transition temperature and the melting temperature.

Graphene/PEKK composite powder samples were obtained by in situ polymerisation. Commercial graphene was satisfactory dispersed in high-performance thermoplastic PEKK polymer to improve its electrical properties. However, higher temperature moulding decreases the graphene performance, whereas lower temperature influences dispersion, tool life, and porosity. Thus, the suitable moulding temperature is a key factor influencing the quality of the electrode bar. The self-prepared Graphene/PEKK composite material is a thermoplastic composite material, which has the characteristics of light weight, high toughness, low energy consumption, corrosion resistance, good formability, and conductivity. In this research, SPS equipment (Figure 3) was used to obtain the different geometric size bar based on the stroke of equipment and the geometric size of the tools.

SPS is a new powder-metallurgy sintering technology in which metals and other powders are placed in a mould made of graphite and other materials. The specific sintering power supply and pressing pressure are applied to the sintered powder by using upper and lower die-punched and electrified electrodes. High-performance materials are produced by discharge activation, thermoplastic deformation, and cooling. Figure 3 shows the SPS device. It primarily includes the following parts: axial pressure device, water-cooled punch electrode, vacuum chamber, atmosphere-control system (vacuum, argon), direct current (DC) pulse and cooling water, displacement measurement, temperature measurement, and safety-control units.

A previous article discussed how to process powder electrodes [43], as well as the factors affecting the moulding quality, including the glass-transition temperature, pre-pressing temperature, heating rate, cooling rate, press times, and other parameters. They directly influence the conductive and mechanics behaviour of self-prepared Graphene/PEKK composites. Accordingly, in the current work, some experiments were designed by the orthogonal analysis method to determine the parameters during the moulding process of self-prepared conductive Graphene/PEKK composite powder. Figure 4 shows scanning electron microscopy (SEM, Hitachi, Tokyo, Japan) analysis self-prepared graphene composite powder and bar. Figure 4a shows the micromorphology of the self-prepared graphene composite powder with 3.8% graphene. Graphene was evenly dispersed into the PEKK matrix by in-situ polymerisation, and the powders were processed into the bar by using the SPS, as shown in Figure 4b,c, which shows the micromorphology of the self-prepared graphene composite powder bar. The graphene sheet layer structure was distributed inside the bar.

To reduce the fitting time and evaluate the computation results, an effective experimental design method was needed. Several different factors and levels needed to be considered when optimizing the parameters during SPS moulding. The purpose of the optimisation of parameters during SPS moulding was to reach the maximum conductivity. Orthogonal design is a kind of design method that is primarily used to study multiple factors and multiple levels [44]. The suitable level range reduces the number of optimisation design alternatives. Therefore, before the orthogonal analysis, the pre-pressure is necessary to predict [45] heating rate, cooling rate, and temperature. Finally, five five-level factors are contained in the orthogonal array used for the optimisation of parameters during SPS moulding, as shown in Table 1. Conductivity tests were conducted on 25 groups of orthogonal experiment samples, and the conclusion that the sample with the process parameters (10, 5, 50, 15, 5) had the highest electrical conductivity was drawn. Thus, the parameters (10, 5, 50, 15, 5) can be used as parameters for a reasonable process for making electrodes with 3.8% graphene as shown in Table 2.

Five electrode bars were obtained based on the parameters in Table 2. The reasonable process parameters were determined by orthogonal analysis. Graphite with good thermal conductivity was used as the mould of the equipment. After moulding, the unloading work was completed, and the graphite paper was used to wrap the Graphene/PEKK composite. The high pressure was 10 MPa, and it quickly reached 10 MPa. The pre-pressure duration was 5 min, over which time the temperature was increased to 145 °C at a rate of 50 °C/min (holding time was 5 min).

Figure 5 shows the electrical conductivity of the Graphene/PEKK bar distribution for different heights. When 0.5 mm of the the electrode bar was polished off in the axial direction, the electrical conductivity was 0.0135 S/cm. With increased grinding depth, the electrical conductivity was almost maintained at the same level. Thus, a content of 3.8% for graphene in the Graphene/PEKK composite powder can be regarded as the conducting graphene composite electrode.

Graphene/PEKK composite with 3.8% graphene content had strong electrical conductivity by using the SPS moulding bar; Figure 6a shows the micromorphology of self-prepared pure PEEK powder, and Figure 6b shows the micromorphology of the self-prepared Graphene/PEKK composite bar. Figure 6 shows the microstructure of the Graphene/PEKK composite compared with the PEKK polymer, which clearly shows that the self-prepared Graphene/PEKK can overcome the phenomenon of graphene agglomeration. Moreover, the material had better fluidity and dispersion at 145 °C to form electrode rods with better conductive properties.

The above research confirmed the electrical conductivity and matrix structure of self-prepared Graphene/PEKK composites and some shortcomings about SPS moulding technology. Compared with hot-press-moulding SPS moulding technology, it can save time as it needed only 30 min. Normally, the temperature can change the grain size. In particular, high temperature can produce toxic gases and influence the molecular structure, whereas low temperature sintering can affect the density of specimens. In other words, it influenced electrical conductivity and strength. At the same time, during SPS, the geometrical properties of the tool limited the specimen size.

Moulding pressure directly affected the density, and at the same time increased the hardness and electrical conductivity by increasing the pressure; increasing the temperature can reduce the flow resistance and shear force between the material particles, as well as reduce the friction coefficient between particles to overcome mould cracks. Although increased temperature reduced the external force, it also reduced the mechanical properties of the material. Therefore, balancing the moulding pressure and moulding temperature was the key to mould polymer composites. On the basis of the above research, we have reason to believe that the product of conducting Graphene/PEKK composite can be obtained by electromagnetic pulse compaction. Accordingly, the following research is discussed.

## 3. Electromagnetic Pulse Compaction

### 3.1. Experimental Equipment

Electromagnetic pulse processing equipment is a new technology equipment which uses electromagnetic energy to process metal materials and is applicable in the field of special processing technology. It is especially suitable for processing non-ferrous materials, such as aluminium, magnesium, and copper. It can weld all kinds of homogeneous or heterogeneous metal materials with conductive properties, which cannot be achieved by traditional technology.

Herein, electromagnetic moulding experiments were performed with a machine, and the technological parameters are given in Table 3. The self-prepared Graphene/PEKK composite powder was processed into the electrode bar. To decrease the cost and determine the process parameters, FE analysis was proposed in this paper. Previous studies have also focused on the electromagnetic pulse compaction of different powder materials, which can reduce the adverse effects of composition segregation and grain growth. Thus, it has wide-ranging application prospects in nanopowder moulding.

The electromagnetic moulding powder pressing machine used is shown in Figure 7. Figure 7 shows the axial compaction process of the plate coil. The working coil was connected with the discharge capacitor of the electromagnetic moulding equipment. The capacitor discharged the working coil, and the impulse current flowed into the working coil. The driving wire pushed the amplifier together under the force of the impulsive magnetic field. The movement was downwards to achieve the compaction of Graphene/PEKK composite powder. The electromagnetic moulding powder compactor enabled high-energy-rate impact compaction, which can increase the powder density, cemented carbide, and magnet powder by 20 to 50% compared with ordinary compaction, which is suitable for the high-density compaction of nanomaterials.

Graphene composite materials have high electrical conductivity, and local welding can be achieved during the process of electromagnetic pulse compaction. The electromagnetic pulse compaction of Graphene/PEKK composite materials was primarily divided into four stages as shown in Appendix A provided the electronic supporting information file in the Appendix A Section. In the first stage, the powder was placed in the mould at a 1:6 proportion, and the powder was loosely stacked. In the second stage, the punch compacted the loose powder and formed a preliminary arch bridge hole between the powders. In the third stage, the magnetic pulse compacted the powder at high speed. The required time was short. A large amount of heat originated from the particle surface. A high temperature and high pressure, adiabatic gas formed in the gap. With increased magnetic pulse pressure, the powder particles underwent elastoplastic deformation, and the contact area increased; thermal energy rapidly dispersed the temperature and melted some small particles. When the magnetic pulse pressure exceeded the limit strength of the powder particles, the powder was destroyed within a very short time to increase the contact area, and welding further proceeded under high temperature and high pressure to achieve the final shape. Lastly, when the pulse was over and the punch was reset, the sample was obtained within a few microseconds. In summary, on the basis of the work principle and process parameters of the experimental equipment, the finite element numerical analysis was explored in the following section.

### 3.2. Finite Element Analysis

#### 3.2.1. Electrical and Magnetic Analysis

During the electromagnetic-moulding process, the energy-conversion process was more complicated, and the mutual coupling of complex energy fields realized powder moulding. Previous research focused on the electromagnetic moulding of metal powder, whereas self-prepared Graphene/PEKK composite material with high electrical conductivity was studied in the current work. Theoretically, the moulding mechanism of a polymer composite powder is similar to that of electromagnetically formed metal powder. This research firstly determined the process parameters of electromagnetically formed conductive Graphene/PEKK composite powder through FE simulation analysis; that provided basic guidance for future experiments. This research used a loosely coupled method with the ANSYS/Multiphysics module to establish the current excitation. The electromagnetic field model was used to study the influence of discharge current on the magnetic field and magnetic pressure with magnetic pressure as the boundary condition in the MSC.Marc software (2019), as shown in Figure 8.

The discharge time of the electromagnetic pulse compaction was shorter than the compaction moulding time. The influence of the elastoplastic deformation of powder on the magnetic-field distribution and the magnetic force was ignored. The FE model was established by the loose coupling method [22]. Firstly, electromagnetic-field analysis was performed, and then powder-moulding analysis was conducted with the electromagnetic-field solution process as the boundary condition. According to the inherent adjustable range of the experimental equipment, four fixed parameters of discharge pressure (1000, 2000, 3000, and 5000 V) and discharge capacitance (1000, 3500, 5000, and 8200 µF) were selected. According to the different factors and levels, an experimental plan was designed, as shown in Table 4. In fact, during the electromagnetic process, energy was transmitted.

Table 4 shows that with increased discharge capacitance, the discharge voltage, discharge current, and magnetic pressure all increased, as shown in Figure 9a,b. However, the magnetic pressure was negative. The same conclusion was reached with increased discharge voltage, the discharge capacitance and magnetic force increased. However, the magnetic force was negative, as shown in Appendix A provided in the electronic supporting information file. Additionally, with increased energy, discharge current and magnetic force increased, but the magnetic force was negative, as shown in Appendix A provided the electronic supporting information file.

In fact, during the electromagnetic pulse compaction processing, the electromagnetic force and discharge current had a pulse signal, as shown in Figure 10 and Figure 11, indicating the different voltage influences on the discharge current and the different magnetic force influences (Appendix A provided in the electronic supporting information file show the different discharge capacitance influences on the discharge current and the different electromagnetic force). The different voltage influence on electromagnetic force respectively. At the same time, in the same discharge cycle, when the discharge current reached the peak value, the electromagnetic pressure also reached the maximum. In other words, the maximum discharge-current influence on the maximum magnetic force was obtained by adjusting the capacitance and voltage. Only a few microseconds were needed to complete the electromagnetic-force reserve.

#### 3.2.2. Moulding Analysis

Electromagnetic pulse compaction can achieve the parameters of the macroscopic powder-compaction process through experimental methods, but it cannot accurately describe the powder-compaction process. This research used a loosely coupled method with the ANSYS/Multiphysics module to establish current excitation. The electromagnetic-field model was used to study the influence of discharge current on magnetic field and magnetic pressure. With magnetic pressure as the boundary condition, in MSC.MARC, the powder module and Shima–Oyane yield criterion were used to establish the powder-compaction model.

Yield criterion: Shima–Oyane [46]
(1)$$F=\frac{1}{\gamma }{\left(\frac{3}{2}\sigma^{d}\sigma^{d}+\frac{\sigma_{m}{}^{2}}{\beta^{2}}\right)}^{0.5}-\sigma_{y}$$
where $\sigma_{y}$ is the uniaxial yield stress, $\sigma^{d}$ is the deviator stress tensor, $\sigma_{m}$ is the hydrostatic pressure, γ, β are the material parameters, $\sigma_{y}$ is a function of temperature and relative density, and γ, β is a function of relative density.
(2)$$\gamma ={\left(b_{1}+b_{2}\rho^{b_{3}}\right)}^{b_{4}}$$
(3)$$\beta ={\left(q_{1}+q_{2}\rho^{q_{3}}\right)}^{q_{4}}$$
where ρ is the relative density.

To reduce the amount of calculation and save time, this study used 1/16 of the sample as the geometric model of the simulation analysis, as shown in Figure 12. During energy release, each node of the electrode bar was uniformly stressed. Self-prepared Graphene/PEKK composite powder samples with 3.8% graphene were produced by the electromagnetic pulse compaction process. The specimens were 40 mm in height and 20 mm in diameter. Although a plane-stress 2D model is sufficient to describe the moulding process and saves computational resources, a 3D model is much easier to understand and use to discuss the joint mechanism of the moulding process.

In the current research, the magnetic-pressure distribution on the bar was obtained using ANSYS. Magnetic pressure was used as the boundary condition of electromagnetic moulding to realize FE numerical simulation analysis. Figure 13 shows the contour of the density distribution on the bar when the discharge voltage was equal to 5000 V for different discharge capacitance values (1000, 3500, 6000, and 8200 µF). The density was the maximum (0.9895) at the edge of the bar. Obviously, the corresponding electromagnetic force increased linearly with increased voltage and current capacitance.Figure 13 shows the influence of different energies on compaction densification (*E* = 0.5 × *CV*^2^; C-capacitance; V-voltage). With increased discharge energy, densification also increased as shown in Appendix A, provided in the electronic supporting information file.

In general, through the simulation analysis of the electromagnetic pulse compacted Graphene/PEKK composite powder, determining the change trend of the process parameters during moulding was possible, but there was a problem of stress distribution at each node in the simulation, especially in the electromagnetic distribution. In the decay stage after the maximum pressure, the positions of different nodes rose to different degrees. This finding showed that when the powder was sintered and the discharge was over, the elastoplastic properties of the material itself changed, and the impulse force acquired inertia, the phenomenon of falling after rising. Therefore, in future research, a constitutive model research should be conducted for self-prepared composite-material powder to further optimize the simulation boundary conditions and improve the simulation accuracy.

## 4. Conclusions

On the basis of above research, the following conclusions were obtained.

The spark plasma sintering technique was used to process the self-prepared Graphene/PEKK polymer composite specimens to get the electrical conductivity of Graphene/PEKK composite; the specimen with diameter 12 mm and height 11 mm was polished off 0.5 mm (1.5, 2.5, 3.5, and 4.5 mm) in the axial direction, and the electrical conductivity was about 0.014S/cm. Thus, 3.8% of the content of graphene in the Graphene/PEKK composite can be regarded as the conducting graphene composite electrode.

The moulding of self-prepared Graphene/PEKK polymer composite powder, discharge voltage, discharge capacitance, and material constitutive model of self-prepared Graphene/PEKK composite powder were important factors affecting the moulding quality. According to the inherent adjustable range of the experimental equipment, four fixed parameters of discharge voltage (1000, 2000, 3000, and 5000V) and discharge capacitance (1000, 3500, 5000, and 8200 µF) were selected.

Simulation analysis using ANSYS and MSC.Marc was proposed, and it used ANSYS to obtain the magnetic force. Additionally, the magnetic-force distribution on different points was regarded as the boundary conditions of MSC.Marc software. It obtained 16 groups of simulation results regarding the discharge current and magnetic force. The different stresses, strains, and density distributions on the electrode bar were determined by simulation analysis. When the energy was the maximum, the stress, strain, and density also arrived at the maximum within a few microseconds.

The self-prepared Graphene/PEKK composite powder will be modified and commercialized, so the material constitutive model must be reconstructed, and the existing constitutive model cannot accurately predict stress, strain and compaction density during the moulding process, especially considering the lack of accuracy in the law of stress changes. Subsequently, we will finish the electromagnetic pulse compaction experiment for self-prepared Graphene/PEKK composite powder based on the process parameters supported by finite element numerical analysis.

## Abbreviation


**No**

**Original Words**

**Abbreviation**
1Poly (ether-ketone-ketone)PEKK2Poly (ether-ether-ketone)PEEK3Spark Plasma SinteringSPS4Lead Zirconate TitanatePZT5Magnetic Pulsed CompactionMPC6Dynamic Magnetic ConsolidationDMC7Direct CurrentDC8Differential Scanning CalorimetryDSC9Scanning Electron MicroscopySEM10Finite ElementFE

## Figures and Tables

**Figure 1 materials-14-00636-f001:**
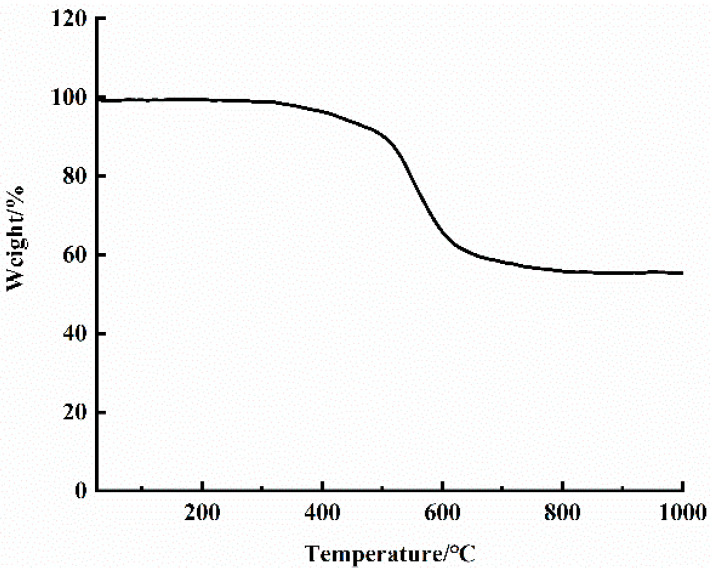
Thermogravimentric analysis (TGA) curve of self-prepared Graphene/PEKK composite powder.

**Figure 2 materials-14-00636-f002:**
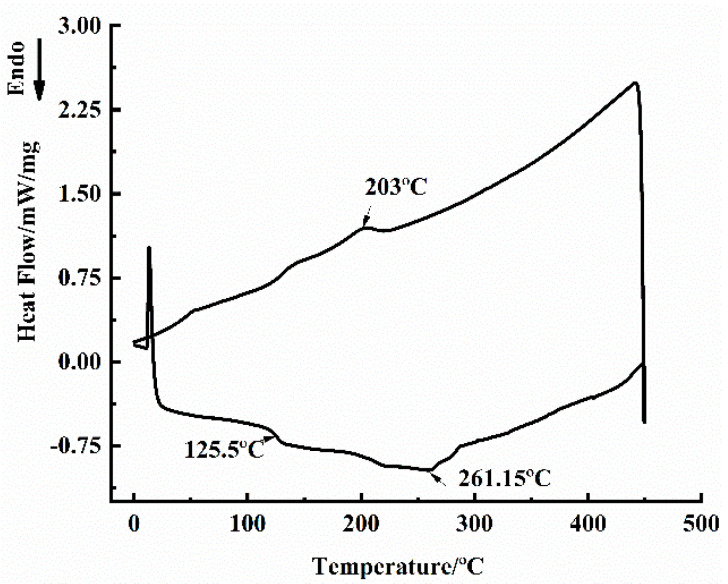
DSC analysis for self-prepared Graphene/PEKK composite powder.

**Figure 3 materials-14-00636-f003:**
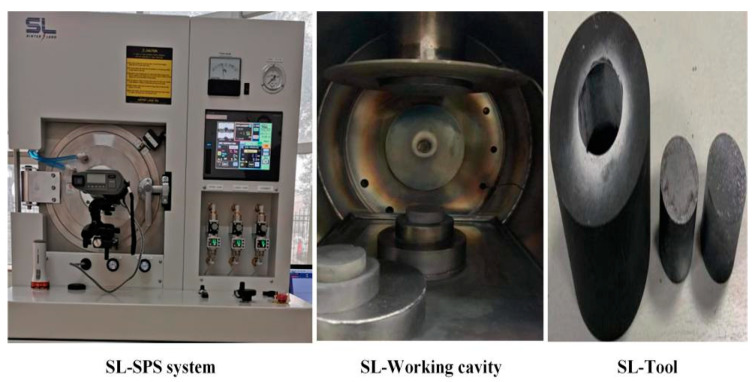
SL-spark plasma sintering (SPS) equipment (Sinter Land Iinc, Niigata Prefecture, Japan).

**Figure 4 materials-14-00636-f004:**
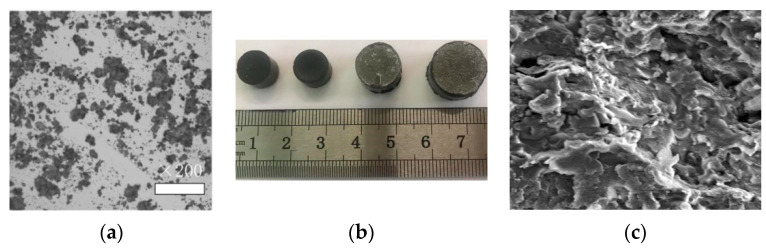
Scanning electron microscopy (SEM) analysis self-prepared graphene composite powder and bar. (**a**) Micro-structure; (**b**) Electrode bar; (**c**) Micro-structure.

**Figure 5 materials-14-00636-f005:**
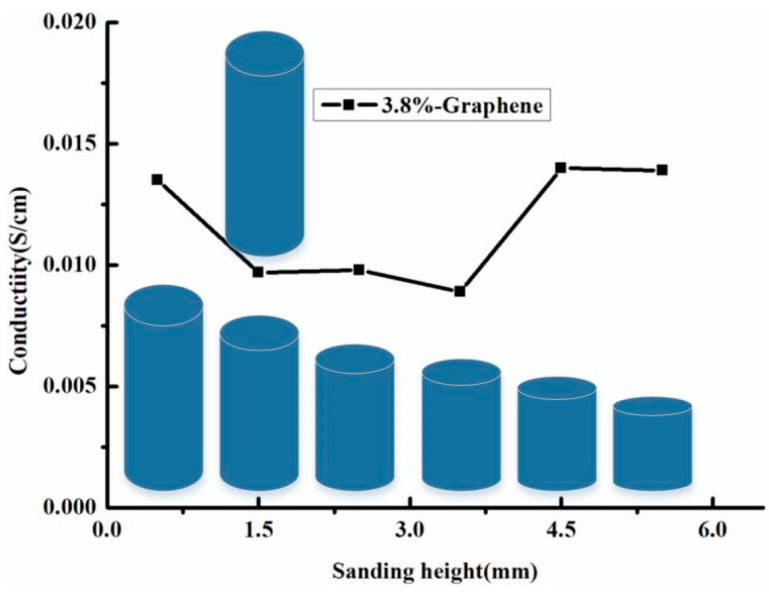
The electrical conductivity of Graphene/PEKK bar distribution on the different height.

**Figure 6 materials-14-00636-f006:**
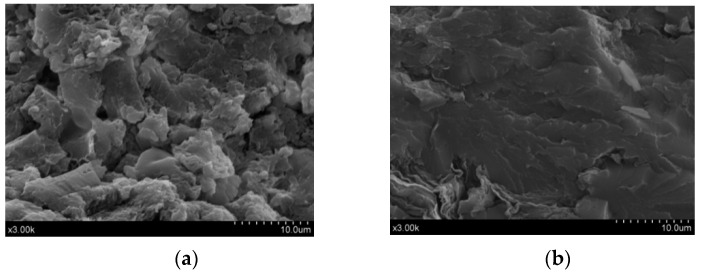
SEM analysis Graphene/PEKK and pure PEKK bar by spark plasma sintering. (**a**) Pure-PEKK; (**b**) Graphene/PEKK composite.

**Figure 7 materials-14-00636-f007:**
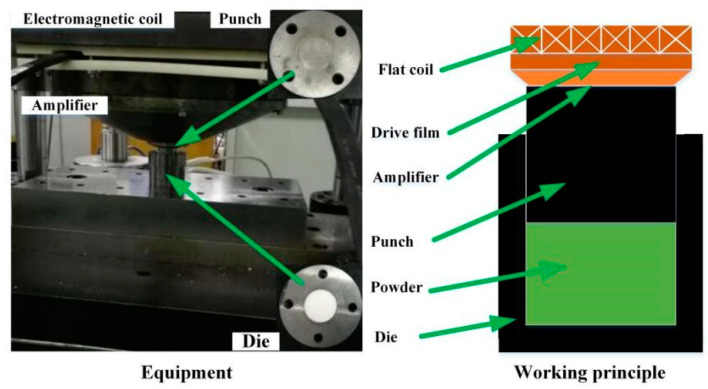
Electromagnetic assisted powder compaction equipment.

**Figure 8 materials-14-00636-f008:**
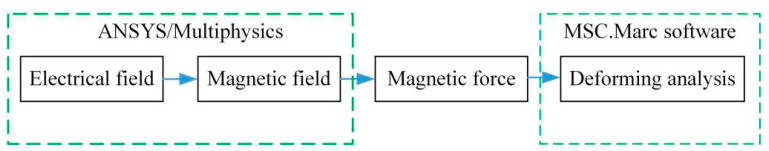
Simulation progress by loosely couple combination of ANSYS with MSC.Marc software.

**Figure 9 materials-14-00636-f009:**
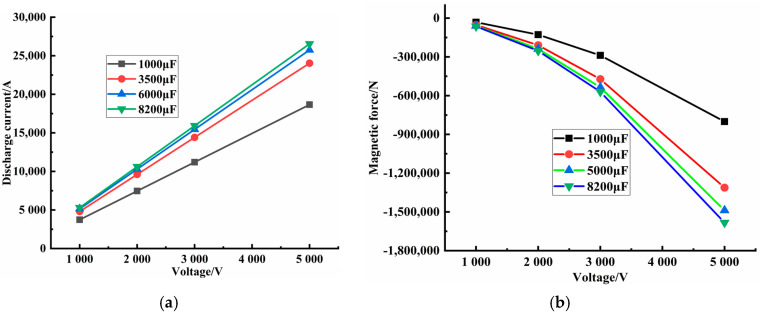
Magnetic force-voltage during the different capacitance. (**a**) Relationship between voltage and current; (**b**) Relationship between voltage and magnetic force.

**Figure 10 materials-14-00636-f010:**
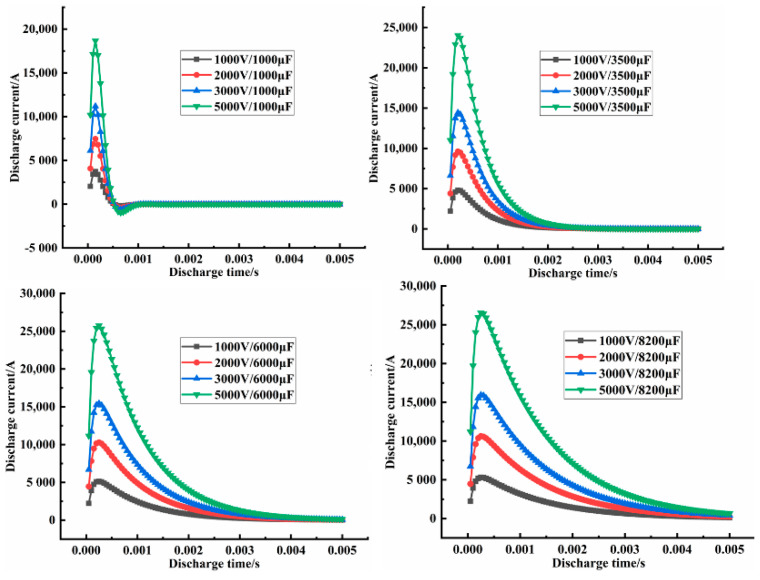
The different voltage influence on discharge current.

**Figure 11 materials-14-00636-f011:**
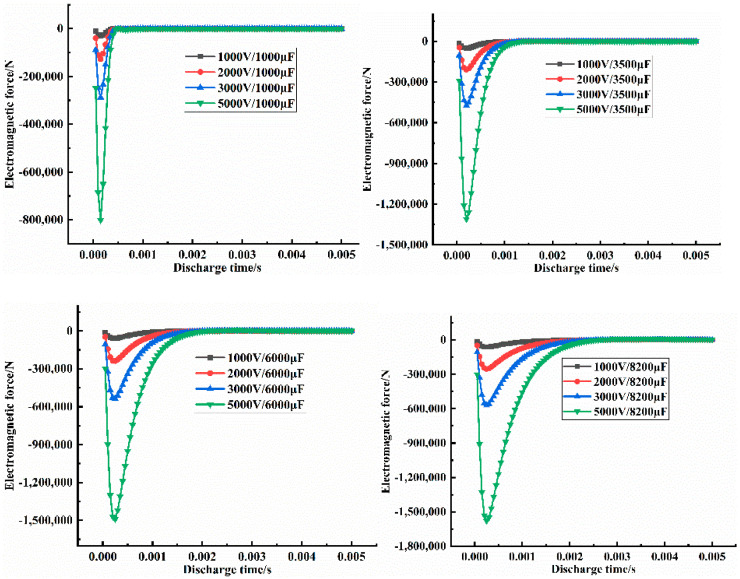
The different voltage influence on electromagnetic force.

**Figure 12 materials-14-00636-f012:**
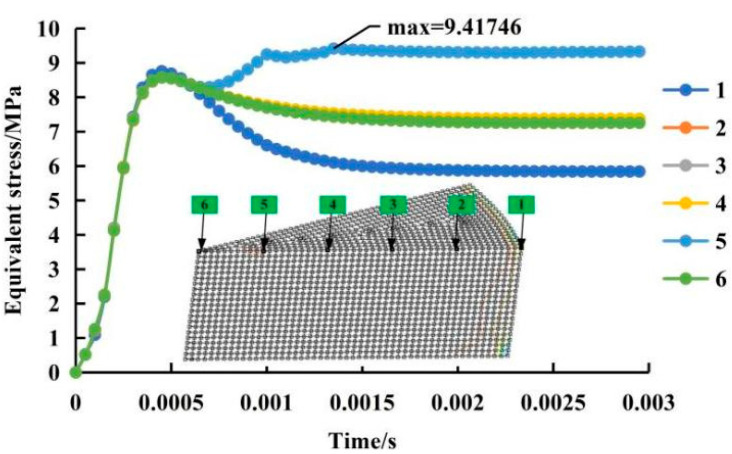
The stress distribution on the bar in the radial direction.

**Figure 13 materials-14-00636-f013:**
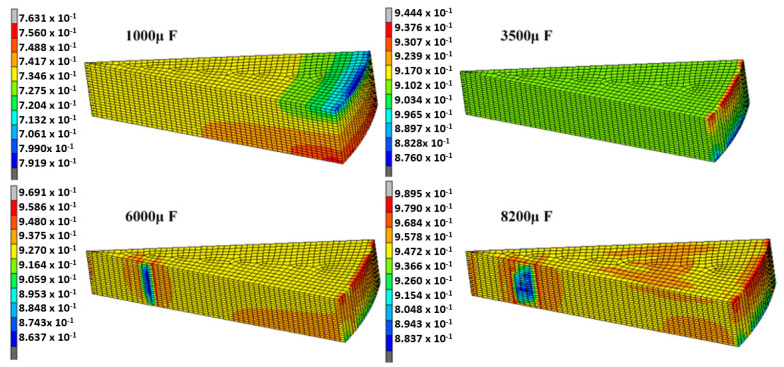
Density distribution on the bar.

**Table 1 materials-14-00636-t001:** The parameters and the levels.

No	Pressure (A)	Pre-Pressure Time (B)	Heating Rate (C)	Cooling Rate (D)	Holding Time (E)
1	6 MPa	1 min	40 °C/min	5 °C/min	1 min
2	8 MPa	3 min	45 °C/min	10 °C/min	3 min
3	10 MPa	5 min	50 °C/min	15 °C/min	5 min
4	12 MPa	7 min	55 °C/min	20 °C/min	7 min
5	14 MPa	9 min	60 °C/min	25 °C/min	9 min

**Table 2 materials-14-00636-t002:** The moulding parameters of the SPS method.

Tools	Pressure	Pre-Pressure	Heating Rate	Cooling Rate	Holding Time	Sintering Temperature	Circulating Water
Graphite	10 MPa	5 min	50 °C/min	15 °C/min	5 min	145 °C	30 °C

**Table 3 materials-14-00636-t003:** The parameters of the electromagnetic pulse compaction device.

Voltage (KV)	Capacitance (μF) (Adjustable)
0.45, 1, 2, 3, 5, 10, 20, 30, 50, 100, 200	500–10,000 μF

**Table 4 materials-14-00636-t004:** Experimental plan and results.

No	Voltage (V)	Capacitance (µF)	Energy (KJ)	Inductance (µH)	Resistance (Ω)	Densification (Max)	Electromagnetic Force (N)	Discharge Current (A)
1	1000	1000	0.5	0.164	13.795	0.4286	−32,020	3732.2
2	1000	3500	1.75	0.164	13.795	0.4796	−52,480	4805.96
3	1000	6000	3	0.164	13.795	0.5147	−59,500	5148.8
4	1000	8200	4.1	0.164	13.795	0.5774	−63,340	5309.63
5	2000	1000	2	0.164	13.795	0.4808	−128,100	7464.4
6	2000	3500	7	0.164	13.795	0.5911	−209,900	9611.92
7	2000	6000	12	0.164	13.795	0.6578	−238,000	10,297.6
8	2000	8200	16.4	0.164	13.795	0.7614	−253,300	10,619.3
9	3000	1000	4.5	0.164	13.795	0.5383	−288,200	11,196.6
10	3000	3500	15.75	0.164	13.795	0.6962	−472,300	14,417.9
11	3000	6000	27	0.164	13.795	0.7786	−535,500	15,446.4
12	3000	8200	36.9	0.164	13.795	0.961	−570,000	15,928.9
13	5000	1000	12.5	0.164	13.795	0.7631	−800,600	18,661
14	5000	3500	43.75	0.164	13.795	0.9444	−1,312,000	24,029.8
15	5000	6000	75	0.164	13.795	0.9691	−1,488,000	25,744
16	5000	8200	102.5	0.164	13.795	0.9895	−1,583,000	26,548.1

## Data Availability

The data used to support the findings of this study are available from the corresponding author upon request.

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
