# Peer review of "Investigation of Electromagnetic Pulse Compaction on Conducting Graphene/PEKK Composite Powder"

_materials, 2021, doi:10.3390/ma14030636_

Round 1

Reviewer 1 Report

This paper can be published after addressing the following issues:

  • It would be advisable to include a glossary of abbreviation.
  • The experimental section is not clearly included: It is advisable to write a section called “Experimental methods” where all the details are described for both experimental and calculation details (FEM).
  • In general, the paper should be shortened for the sake of the readership. There are several Figures which can be moved to an electronic supporting information.
  • The scientific discussion should be improved and better exposed showing what are the main results and contributions of the research. Please, also improve the abstract and the conclusions by shortening them and highlighting the main contributions.

Author Response

Response to Decision Letter

RE: Manuscript ID: Materials-1070610

Dear editor:

We would like to thank Materials for giving us the opportunity to revise our manuscript. We thank the reviewers for their careful read and thoughtful comments on previous draft. We have carefully taken their comments into consideration in preparing our revision, which has resulted in a paper that is clearer, more compelling and broader. The following summarizes how we responded to reviewer comments. We also appreciated the constructive criticism and suggestion. We addressed all the points raised by the reviewer, as summarized below.

Reviewer 1:

  • It would be advisable to include a glossary of abbreviation.

Answer: On the basis of your suggestion, we have added a glossary of abbreviation as shown in the following.

No

Original words

Abbreviation

1

Poly (ether-ketone-ketone)

PEKK

2

Poly (ether-ether-ketone)

PEEK

3

Spark Plasma Sintering

SPS

4

Lead Zirconate Titanate

PZT

5

Magnetic Pulsed Compaction

MPC

6

Dynamic Magnetic Consolidation

DMC

7

Direct Current

DC

8

Differential Scanning Calorimetry

DSC

9

Scanning Electron Microscope

SEM

10

Finite Element

FE

  • The experimental section is not clearly included: It is advisable to write a section called “Experimental methods” where all the details are described for both experimental and calculation details (FEM).

Answer: According to your suggestion, we have revised Section 2 and Section 3 as shown in the body of the manuscript. At the same time, we also planned the main content in the manuscript. Section 2 showed the performance of self-prepared Graphene/PEKK composite, including (1) Graphene/PEKK composite powder was synthesized based on in-situ polymerization method in our lab, (2) The different geometrical size bars were obtained by the SPS, (3) The conductivity of the bars were measured in order to determine the feasibility through the electromagnetic pulse compaction for the Graphene/PEKK composite powder. Section 3 showed the electromagnetic pulse compaction, including (1) The work principle of electromagnetic pulse compaction equipment and process parameters, (2) On the basis the work principle and process parameters of electromagnetic pulse compaction equipment, finite element numerical analysis was designed, (3) Electrical and magnetic analysis, (4) Moulding analysis for Graphene/PEKK composite powder.

  • In general, the paper should be shortened for the sake of the readership. There are several figures which can be moved to an electronic supporting information.

Answer: On the basis of your suggestion, we have shortened for the sake of the readership as shown in the body of the manuscript. In the original paper, Figure.6, Figure.7, Figure.9, Figure.10, Figure.12, Figure.16 have be moved to an electronic supporting information, at the same time, we also showed DSC curve as shown in the body of the manuscript.

  • The scientific discussion should be improved and better exposed showing what are the main results and contributions of the research. Please, also improve the abstract and the conclusions by shortening them and highlighting the main contributions.

Answer: According to your suggestion, we have revised the main results and contributions of the research, and then we also improved the abstract and the conclusions by shortening them and highlighting the main contributions as shown in the body of the manuscript.

Please see the detailed information in the revised manuscript, and the modified contexts are highlighted by yellow.

Thanks for all the help. Please do not hesitate to send us emails if you have any more questions or concerns.

Thanks for all the help.

Best wishes,

Professor Dr, Fan XU

2021.01.14

Reviewer 2 Report

The manuscript reported a molding process of graphene based polymer composite. The composite was obtained using hot molding and cold molding electromagnetic pulse compaction method was proposed too. The proposed molding method was then compared with hot molding (spark plasma sintering). I found the manuscript title is leading to misinterpretation to the readers as a review article (Research progress…). I would suggest changing the title according to the text. The manuscript should be rewritten. It was not clear which molding method should be adopted.

  1. Line 70 says “conductive polymer composites…”. Please clearly indicate what does author meant by this. It is confusing here, as conductive polymers (polyaniline, polypyrrole etc.) are different set of polymers.
  2. Please check line 85. It does not make any sense. Also, please specify what all the acronyms stands for wherever they have used or mention them when they are used for the first time in the text.
  3. What does author meant by self-prepared graphene/PEKK composite? It is not clear.
  4. Line 181 says “SPS equipment was used to obtain the different geometrical bar.” The figure shows only different sizes cylindrical shape. Please show other shapes as well.
  5. Author states that the melting temperature was obtained from DSC method. Please include DSC curve for the same.
  6. Line 166, 316 “what conductivity was high; thermal or electrical?
  7. What are the mechanical properties of obtained composite? How do you compare those if composite was obtained using proposed method?

Author Response

Response to Decision Letter

RE: Manuscript ID: Materials-1070610

Dear editor:

We would like to thank Materials for giving us the opportunity to revise our manuscript. We thank the reviewers for their careful read and thoughtful comments on previous draft. We have carefully taken their comments into consideration in preparing our revision, which has resulted in a paper that is clearer, more compelling and broader. The following summarizes how we responded to reviewer comments. We also appreciated the constructive criticism and suggestion. We addressed all the points raised by the reviewer, as summarized below.

Reviewer 2:

Comments to the Author

The manuscript reported a molding process of Graphene based polymer composite. The composite was obtained using hot molding and cold molding electromagnetic pulse compaction method was proposed too. The proposed molding method was then compared with hot molding (Spark Plasma Sintering). I found the manuscript title is leading to misinterpretation to the readers as a review article (Research progress…). I would suggest changing the title according to the text. The manuscript should be rewritten. It was not clear which molding method should be adopted.

  • According to the reviewer’s suggestion, we have revised the title and the body of manuscript as shown in the body of manuscript.

Answer: Title: Investigation of Electromagnetic Pulse Compaction on Conducting Graphene/PEKK Composite Powder

  • Line 70 says “conductive polymer composites…”. Please clearly indicate what does author meant by this. It is confusing here, as conductive polymers (polyaniline, polypyrrole etc.) are different set of polymers.

Answer: What we want to express is conducting composite with a insulating polymer matrix and a conducting filler, such as conducting Graphene/PEKK composite, etc.. Conducting property of polymer composite depends on conducting filler, while conducting property of intrinsic conductive polymer (such as polyaniline, polypyrrole etc.) is determines by its conjugated structure.

According to the suggestion on revision, it has been revised as the following in the revised manuscript: Polymer composites with conducting filler are extensively used in some fields, but numerous fillers are needed to obtain high conductivity, leading to difficulties in processing and embrittlement of materials.

  • Please check line 85. It does not make any sense. Also, please specify what all the acronyms stands for wherever they have used or mention them when they are used for the first time in the text.

Answer: According to your suggestion, we have deleted line 85 (So to select a suitable method can obtain the high quality product). At the same time, we also specified some acronyms as shown in the body of the manuscript.

  • What does author meant by self-prepared graphene/PEKK composite? It is not clear.

Answer: Self-prepared Graphene/PEKK composite powder were synthesized on the basis of in situ polymerization in our Lab. At the same time, Section 2 introduced in detail self-prepared Graphene/PEKK composite as shown in the body of the manuscript.

  • Line 181 says “SPS equipment was used to obtain the different geometrical bar.” The figure shows only different sizes cylindrical shape. Please show other shapes as well.

Answer: According to your suggestion, we have revised “SPS equipment was used to obtain the different geometrical size bar” as shown in the body of the manuscript (We only processed the different size bars). The stroke of equipment can change the height of the specimen. The diameter of the tools can determine the diameter of the specimen.

  • Author states that the melting temperature was obtained from DSC method. Please include DSC curve for the same.

Answer: On the basis of your suggestion, DSC curve has been added in the body of the manuscript. The melting temperature was determined by DSC ( Differential Scanning Calorimetry ) test. Figure.2 presents DSC curves of Graphene/PEKK composite with a grapheme mass content of 3.8%. As shown in Figure.2, the Graphene/PEKK composite shows one glass transition and one melting transition. It should be noted that melting temperature and corresponing viscosity of polymer composite directly determines the difficulty and quality of flow forming process in engineering application. According to Figure.2, melting temperature of Graphene/PEKK composite obtained from DSC curve is 261℃. Normally, moulding temperature (sintering temperature) was initially determined between the glass transition temperature and the melting temperature through many experiments.

Figure.2. DSC analysis for self-prepared Graphene/PEKK composite powder

  • Line 166, 316 “what conductivity was high; thermal or electrical?

Answer: Graphene/PEKK composite powder with high conductivity can be processed by  electromagnetic pulse compaction based on the principle of electromagnetic pulse forming for metal powder. So in this paper, conductivity of Graphene/PEKK composite will be regarded as main factor to be discussed.

  • What are the mechanical properties of obtained composite? How do you compare those if composite was obtained using proposed method?

Answer: According to your suggestion, we have revised the manuscript.

  • The previous manuscript was disorganized, so we modified the general structure of the article, especially Section 2 and Section 3.

Section 2 shows the performance of self-prepared Graphene/PEKK composite, including (1) Graphene/PEKK composite powder was synthesized based on in-situ polymerization method in our lab, (2) The different geometrical size bars were obtained by the SPS, (3) The conductivity of the bars were measured in order to determine the feasibility through the electromagnetic pulse compaction for the Graphene/PEKK composite powder. Section 3 shows the electromagnetic pulse compaction, including (1) The work principle of electromagnetic pulse compaction equipment and process parameters, (2) On the basis the work principle and process parameters of electromagnetic pulse compaction equipment, finite element numerical analysis was designed, (3) Electrical and magnetic analysis, (4) Moulding analysis for Graphene/PEKK composite powder.

  • The title “Research Progress of Conducting Graphene Polymer Composite PowderMoulding Process”has been revised to “Investigation of Electromagnetic Pulse Compaction on Conducting Graphene/PEKK Composite Powder”. At the same time, we have revised the abstract, body, conclusions and other parts.
  • In our research, the electromagnetic pulse compaction will be used to mould conducting Graphene/PEKK composite powder at room temperature. The Graphene/PEKK composite powder was processedby Spark Plasma Sintering(SPS) to determine the conductivity of Graphene/PEKK composite, and then it will further determine that conductive Graphene/PEKK composite powder can be processed by electromagnetic pulse forming technology.
  • Graphene/PEKK composite powder weresynthesized in our Lab according toin-situ polymerization. Electromagnetic pulse compaction will be used to mould conducting Graphene/PEKK composite powder at room temperature. In the future, self-prepared Graphene/PEKK composite powder will be processed the electrode bar to monitor flow of the oil field exploitation underground. So, we will try to use the electromagnetic pulse forming technique to mould the Graphene/PEKK composite powder. Currently, the self-prepared Graphene/PEKK composite powder are not commercialized materials. According to the working principle of electromagnetic forming conducting powder, we attempted to carry out Graphene/PEKK composite powder research. In order to reduce the loss of laboratory materials, it is essential to determine the distribution of stress, strain and density through finite element analysis method.
  • This research used a loosely coupled method with the ANSYS/Multiphysics module to establish the current excitation. The electromagnetic field model was used to study the influence of discharge current on magnetic field and magnetic pressure with magnetic pressure as the boundary condition in Marc software, as shown in Figure.7.

Figure.7. Simulation progress by loose couple combine ANSYS with MSC.Marc software.

  • Subsequently, we will finish the electromagnetic pulse compaction experiment for self-prepared Graphene/PEKK composite powder based onthe process parameterssupported by finite element numerical analysis.

Please see the detailed information in the revised manuscript, and the modified contexts are highlighted by yellow.

Thanks for all the help. Please do not hesitate to send us emails if you have any more questions or concerns.

Thanks for all the help.

Best wishes,

Professor Dr, Fan XU

2021.01.14

Round 2

Reviewer 2 Report

Author certainly made the required changes in the manuscript. However, there are few minor issues that need to be addressed before its acceptance.

  1. Please clearly indicate the conductivity being discussed: thermal or electrical. Some statements state electrical conductivity, but some does not. As the discussion also talks about molding temperature, it should be clearly mentioned; is it thermal or electrical conductivity that's being discussed.
  2. Author are suggested to cite all the figures in the text including from the supporting information provided. Also, cite tables in the text as table instead of tab.
  3. The DSC curve showed the just melting temperature of the composite, it would be better if curve shows other parameter obtained (such as Tg/Tcrys) from it and was labelled with exothermic and endothermic direction (up or down). Also, author is suggested to include TGA analysis for better comparison. Please include experimental section and procedure for thermal analyses in the text.

Author Response

Response to Decision Letter

RE: Manuscript ID: Materials-1070610

Dear editor:

We would like to thank Materials for giving us the opportunity to revise our manuscript. We thank the reviewers for their careful read and thoughtful comments on previous draft. We have carefully taken their comments into consideration in preparing our revision, which has resulted in a paper that is clearer, more compelling and broader. The following summarizes how we responded to reviewer comments. We also appreciated the constructive criticism and suggestion. We addressed all the points raised by the reviewer, as summarized below.

Reviewer 2:

Comments to the Author

Author certainly made the required changes in the manuscript. However, there are few minor issues that need to be addressed before its acceptance.

  • Please clearly indicate the conductivity being discussed: thermal or electrical. Some statements state electrical conductivity, but some does not. As the discussion also talks about molding temperature, it should be clearly mentioned; is it thermal or electrical conductivity that's being discussed.

Answer: According to your suggestion, we have revised this question as shown in the body of the manuscript, and the modified contexts are highlighted by green.

Line 13, 17, 166, 284, 286, 291, 292, 303, 309, 312, 375, 481, 484

  • Author are suggested to cite all the figures in the text including from the supporting information provided. Also, cite tables in the text as table instead of tab.

Answer: According to your suggestion, we have revised this question as shown in the body of the manuscript, and the modified contexts are highlighted by red.

Line 195, 196, 206, 207, 208, 216, 221, 231, 234, 240, 251, 252, 255, 260, 270, 274, 277, 284, 291, 293, 294, 295, 302, 329, 331, 338, 339, 349, 354, 383, 385, 395, 397, 401, 402, 404, 405, 409, 411, 412, 413, 414, 423, 426, 445, 453, 456, 461, 463, 464, 466.

  • The DSC curve showed the just melting temperature of the composite, it would be better if curve shows other parameter obtained (such as Tg/Tcrys) from it and was labelled with exothermic and endothermic direction (up or down). Also, author is suggested to include TGA analysis for better comparison. Please include experimental section and procedure for thermal analyses in the text.

Answer: On the basis of your suggestion, thermal analysis including TGA test and DSC test have been added in the body of the manuscript, respectively.

Thermal properties including Thermogravimentric analysis (TGA) and Differential scanning calorimetry (DSC) of self-prepared Graphene/PEKK composite powder were characterized before moulding of conductive Graphene/PEKK composite materials. TGA test (TGA-NETZSCH STA 449F3, Germany) was conducted from 50 to 1000 ℃ at a ramp rate of 10 ℃/min. DSC test (DSC- NETZSCH STA 449F3, Germany) was carried out between 0 and 450 at a heating /cooling rate of 10 ℃/min  

Figure 1 shows TGA curve of Graphene/PEKK composite with a grapheme mass content of 3.8%. As is shown in Figure 1, self-prepared Graphene/PEKK composite exhibits an one-step weight loss process, which indicates good physical compatibility between Graphene and PEKK matrix in the composite. A close look reveals that there was virtually almost no weight loss before 300℃, and eight loss of Graphene/PEKK composite is around 5% when temperature increased to 450℃, which can be defined as the start decomposition temperature. Main decomposition of the Graphene/PEKK composite occurs around 561℃ with a weight loss of 25%. It should be noted that the weight loss of the Graphene/PEKK composite is 45% when temperature is around 1000℃. The results indicated high thermal stability of Graphene/PEKK composite.

Figure1. TGA curve of self-prepared Graphene/PEKK composite powder

    Figure 2 presents DSC curves of Graphene/PEKK composite with a grapheme mass content of 3.8%. As shown in Figure 2, two types of transitions occur in this Graphene/PEKK composite during thermal cycling between 20 and 288℃. The glass transition occurs mainly within the temperature range 114-129℃, whereas the melting and crystallization transitions occur around at 261℃ and 203℃ upon heating and cooling, respectively. The glass transition temperature (Tg) of this the Graphene/PEKK composite may be defined as 125.5℃. It should be noted that melting temperature and corresponding viscosity of polymer composite directly determines the difficulty and quality of flow forming process in engineering application. According to Figure 2, melting temperature of Graphene/PEKK composite obtained from DSC curve is 261℃. Normally, moulding temperature (sintering temperature) was initially determined between the glass transition temperature and the melting temperature.

Figure2. DSC analysis for self-prepared Graphene/PEKK composite powder

Please see the detailed information in the revised manuscript, and the modified contexts are highlighted by yellow.

Thanks for all the help. Please do not hesitate to send us emails if you have any more questions or concerns.

Thanks for all the help.

Best wishes,

Professor Dr, Fan XU

2021.01.21
